# Canadian Collaboration to Identify a Minimum Dataset for Antimicrobial Use Surveillance for Policy and Intervention Development across Food Animal Sectors

**DOI:** 10.3390/antibiotics11020226

**Published:** 2022-02-10

**Authors:** David F. Léger, Maureen E. C. Anderson, François D. Bédard, Theresa Burns, Carolee A. Carson, Anne E. Deckert, Sheryl P. Gow, Cheryl James, Xian-Zhi Li, Michael Ott, Agnes Agunos

**Affiliations:** 1Center for Foodborne, Environmental and Zoonotic Infectious Diseases, Public Health Agency of Canada, Guelph, ON N1H 7M7, Canada; david.leger@phac-aspc.gc.ca (D.F.L.); carolee.carson@phac-aspc.gc.ca (C.A.C.); anne.deckert@phac-aspc.gc.ca (A.E.D.); 2Veterinary Science Unit, Ontario Ministry of Agriculture, Food and Rural Affairs, Guelph, ON N1G 4Y2, Canada; Maureen.E.C.Anderson@ontario.ca; 3Animal Industry Division, Agriculture and Agri-Food Canada, Ottawa, ON K1A 0C5, Canada; francois.bedard@agr.gc.ca; 4Canadian Animal Health Surveillance System, Animal Health Canada, Elora, ON N0B 1S0, Canada; tburns@ahwcouncil.ca; 5Center for Foodborne, Environmental and Zoonotic Infectious Diseases, Public Health Agency of Canada, Saskatoon, SK S7N 5B4, Canada; sheryl.gow@phac-aspc.gc.ca; 6Animal Industry Division, Canadian Food Inspection Agency, Ottawa, ON K1A 0Y9, Canada; cheryl.james@inspection.gc.ca; 7Veterinary Drugs Directorate, Health Canada, Ottawa, ON K1A 0K9, Canada; Xianzhi.li@hc-sc.gc.ca; 8Aquatic Ecosystems Sector, Fisheries and Oceans Canada, Ottawa, ON K1A 0E6, Canada; michael.ott@dfo-mpo.gc.ca

**Keywords:** antimicrobials, multisectoral, stewardship, surveillance, collaboration, antimicrobial use, antimicrobial resistance, food animals

## Abstract

Surveillance of antimicrobial use (AMU) and antimicrobial resistance (AMR) is a core component of the 2017 Pan-Canadian Framework for Action. There are existing AMU and AMR surveillance systems in Canada, but some stakeholders are interested in developing their own AMU monitoring/surveillance systems. It was recognized that the establishment of core (minimum) AMU data elements, as is necessary for policy or intervention development, would inform the development of practical and sustainable AMU surveillance capacity across food animal sectors in Canada. The Canadian Animal Health Surveillance System (CAHSS) AMU Network was established as a multisectoral working group to explore the possibility of harmonizing data inputs and outputs. There was a consensus that a minimum AMU dataset for AMU surveillance (MDS-AMU-surv) should be developed to guide interested parties in initiating AMU data collection. This multisectoral collaboration is an example of how consultative consensus building across relevant sectors can contribute to the development of harmonized approaches to AMU data collection and reporting and ultimately improve AMU stewardship. The MDS-AMU-surv could be used as a starting point for the progressive development or strengthening of AMU surveillance programs, and the collaborative work could serve as a model for addressing AMR and other shared threats at the human–animal–environment interface.

## 1. Introduction

Surveillance is a vital component of the Global Action Plan for antimicrobial resistance (AMR) published by the World Health Organization (WHO) [1], the Food and Agriculture Organization of the United Nations’ (FAO) action plan for AMR 2021-2025 [2] and its predecessor [3], and the World Organisation for Animal Health’s (OIE) strategy for AMR and the prudent use of antimicrobials [4]. The FAO-OIE-WHO tripartite collaboration is one example of how resources and expertise can be optimized to implement actions toward mitigating AMR. Surveillance of antimicrobial use (AMU) and AMR is a component of the Canadian plan for “*Tackling Antimicrobial Resistance and Antimicrobial Use: A Pan-Canadian Framework for Action*” [5]. Surveillance data on AMU and AMR are essential for informed decision making to direct other components of the action plan being developed based on this framework, including infection prevention and control, stewardship, and research and innovation. In Canada, AMU/AMR data directly target AMU interventions and have utility in monitoring the impact of regulatory or voluntary changes in AMU practices.

Currently, there are two official data sources that provide information on the types and quantities of antimicrobials intended for use in animals in Canada. The first is the Veterinary Antimicrobial Sales Reporting (VASR) system, in which manufacturers, importers, and compounders are required to submit their annual sales data for medically important antimicrobials to the Veterinary Drugs Directorate at Health Canada [6]. Reporting the sale of medically important antimicrobials [7] is in compliance with the Regulations Amending the Food and Drug Regulations (Veterinary Drugs–Antimicrobial Resistance) [8]. The data providers are also required to include estimates of sales by animal species. The VASR data collection came into effect in 2018, and the data are incorporated in the Canadian Integrated Program for Antimicrobial Resistance Surveillance (CIPARS), otherwise known as the CIPARS VASR component (i.e., pertains to the CIPARS’s surveillance component, which conducts the data analysis output, reporting, and communication of the antimicrobial sales data collected). Previously, between 2006 and 2018, the annual national antimicrobial sales data were provided to the CIPARS of the Public Health Agency of Canada by the Canadian Animal Health Institute (CAHI) [9,10,11], which did not include estimates of sales by animal species but was rather stratified by companion and production animals (production animals included horses) [11]. The second source of data is the CIPARS Farm AMU/AMR Surveillance program, a voluntary initiative that collects data from a network of sentinel veterinarians and producers in specific livestock sectors (pigs, broiler chickens, turkeys, feedlot beef and currently piloting data collection in dairy cattle and layer chickens) across the country [10]. The two AMU data components of CIPARS described above (1. CIPARS VASR sales data, and 2. CIPARS Farm AMU data) are complementary sources of information that contribute to the general landscape of AMU in animals in Canada. These AMU data are useful to understand the AMR trends and patterns across the food supply chain. In addition to the farm program, beginning in 2017, Fisheries and Ocean Canada (DFO) provided open data on the quantities of antimicrobials used in marine and freshwater finfish aquaculture [10]. These data are part of the reporting requirements by aquaculture industry operators under the Aquaculture Activities Regulations authorized under the Fisheries Act [12].

In 2016, the Canadian Animal Health Surveillance System (CAHSS), now an initiative of Animal Health Canada, formerly the National Farmed Animal Health and Welfare Council (https://animalhealthcanada.ca/, accessed on 1 February 2022), brought together individuals representing various government agencies and private industry organizations to discuss AMU and AMR across animal production sectors in Canada. In this inaugural meeting of the CAHSS AMU/AMR Working Group, it was recognized that protecting the effectiveness of antimicrobials while caring for the health and welfare of Canadian animal populations was a key component of national antimicrobial stewardship. There was also a recognition that policy makers, at various levels, lacked a clear understanding around the data necessary to evaluate AMU and to advance stewardship interventions. To address AMU information gaps and assist interested parties in developing their own AMU surveillance programs, the group determined that the elucidation of a “*minimum data set for AMU surveillance*” (MDS-AMU-surv) was a necessary first step. This core list of AMU variables would contribute to the harmonization of data collection at relevant points in the antimicrobial distribution pathway. 

At the time of the initial meeting in 2016, there was no existing formal One Health governance mechanism for addressing AMR in Canada. However, the CAHSS AMU/AMR Working Group, combined with joint leadership from the Public Health Agency of Canada (CIPARS) and the Canadian Food Inspection Agency, facilitated the discussions to address activities related to the Pan-Canadian Framework for Action on AMR/AMU. Multisectoral collaboration, described as “more than one sector working together (on a joint program or response to an event)” was identified as the key to addressing zoonoses and other shared threats such as AMR in the human–animal–environment interface [13]. This paper aims to describe the AMU distribution pathway for antimicrobials intended for use in animals in Canada, identify points in the distribution chain where data could be collected to describe the multisectoral collaboration that facilitated the development of the MDS-AMU-surv, and to provide examples of the utility of the MDS-AMU-surv for AMU monitoring and research.

## 2. Results

The organizations represented by the CAHSS AMU/AMR are listed in the Appendix A. 

### 2.1. Contextual Review of Existing AMU Surveillance Infrastructures and the Antimicrobial Supply Distribution Pathway in Canada

The key players in the Canadian antimicrobial supply distribution pathway are depicted in Figure 1. The distribution is organized into three levels, which also represent existing and potential AMU data collection points:
(1)Manufacturers, importers, and distributors of veterinary drugs (finished products), importers, manufacturers, and compounders of active pharmaceutical ingredients (APIs) (i.e., those who obtain antimicrobials from pharmacies/dispensers of veterinary drugs and human drugs for animal use (blue line in Figure 1). Reporting to VASR is required by regulation;(2)Prescription and dispensing facilities, including veterinary clinics, feed mills, agri-food companies who employ veterinarians (i.e., corporate veterinarians who prescribe antimicrobials for use in relevant production phases as part of the coordinated supply chain within their network including hatcheries, breeders, commercial growing farms-various commodities, and feedmills), and pharmacies/dispensers of veterinary drugs and human drugs for animal use). Data from this level are not yet available or targeted for a future data collection point;(3)End users of antimicrobials including pet owners and producers of terrestrial and aquatic food animals. Information on specific terrestrial food animal species is captured through the CIPARS Farm AMU/AMR Surveillance program (voluntary) and aquatic animals via the DFO (by regulation).

**Figure 1 antibiotics-11-00226-f001:**
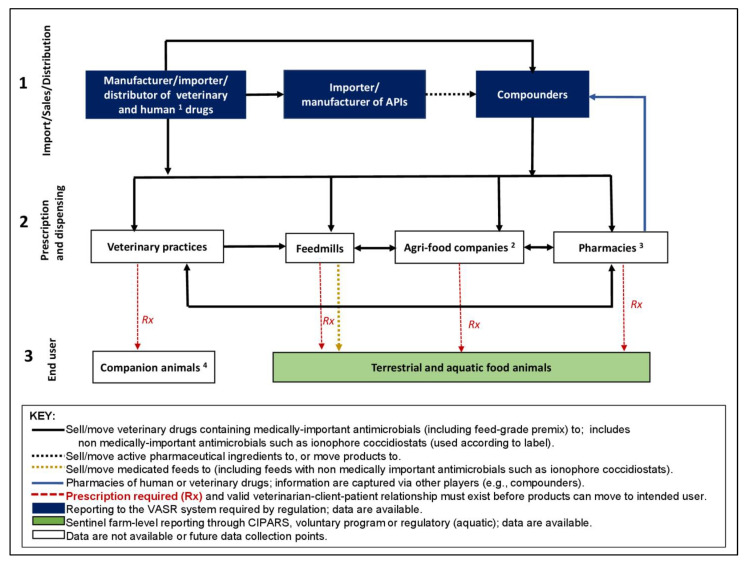
The antimicrobial supply distribution pathway in Canada indicating the existing and future antimicrobial use data collection points. VASR—Veterinary Antimicrobial Sales Reporting system. CIPARS—Canadian Integrated Program for Antimicrobial Resistance Surveillance. ^1^ Human drugs intended for use animals (largely companion animals). ^2^ Veterinarians in agrifood companies (corporate veterinarians) prescribe antimicrobials for use in relevant production phases as part of the coordinated supply chain within their network (e.g., hatcheries, breeders, grower farms, and feedmill). ^3^ Pharmacies of veterinary drugs and human drugs intended for use in animals (i.e., pharmacies of veterinary drugs and human drugs have similar requirements from a regulatory standpoint but differ in where they are dispensing antimicrobials). ^4^ Companion animals include horses.

The report from the Council of Chief Veterinary Officers in Canada on non-human AMU surveillance [14] was a useful resource for the working group. It provides an understanding of the national infrastructure for AMU data collection and the need for AMU surveillance in Canada. Recently, the document was modified to reflect regulatory changes implemented in December of 2018 in how antimicrobial drug products are distributed/sold. In Figure 1, data availabilities are depicted by the colour of the boxes: blue (available), light blue (partial or captured by another player in the distribution chain), or white (no data; future data collection points). Since 2018, regulations have required that manufacturers, importers, and compounders submit annual data on the volume of medically important antimicrobials sold for veterinary use, by animal species, to the VASR system [8]. Manufacturers and importers of veterinary drugs that also sell or move nonmedically important antimicrobials such as ionophores to other actors are not required to report the total volume of nonmedically important drugs at this time. Importers and manufacturers of APIs are required to have a drug establishment license and follow good manufacturing practices [15]. As stated earlier, prior to 2018, antimicrobial sales data were obtained from CAHI, which is a trade association that represents the majority of manufacturers and distributors of animal health products [16]. The CAHI data collection and analysis were performed by a third party (Impact Vet) [17]. Collated CAHI sales data were then voluntarily shared with CIPARS and included in CIPARS annual reports [10].

With the VASR system, data providers electronically enter information into a secure web-based site under the Canadian Network for Public Health Intelligence at the Public Health Agency of Canada [18]. Information is collected on product characteristics (e.g., Health Canada-assigned drug identification number (DIN), product name, package size, number of packages, ATCvet code, antimicrobial active ingredient, class, and formulation), data provider (manufacturer, importer, and compounder), province, intended animal species of use (high and low estimates of packages sold for 11 different animal species groups), quantities sold, and quantities exported. At the time of writing, there was no system for reporting veterinary prescription and dispensation of antimicrobial products in Canada, but a pilot project in select animal sector/volunteer veterinary practices is underway. 

In terms of end user data in terrestrial animals, CIPARS collects farm-level AMU from a network of sentinel veterinarians and producers using species-specific questionnaires or electronic data capture (feedlot beef). The CIPARS program is active in grower–finisher pigs, broiler chickens, turkeys, and feedlot beef with pilot farm-level surveillance in dairy cattle and chicken egg layers, and abattoir-level AMU surveillance (flock sheets) in spent broiler breeder chickens. Farm-level AMU data, animal health, and basic demographics information from the questionnaires are entered into the CIPARS AMU database and exported in Microsoft Excel (Professional Plus 2016) format for further analysis.

A valid veterinarian–client–patient relationship (VCPR) must exist before the finished products containing prescription drugs (including medically important antimicrobials) reach the end user. With the new regulations, veterinary prescriptions and oversight are required for dispensing of medically important antimicrobial products including through feed mills, agrifood companies, and pharmacists. Agrifood companies are similar in structure to vertically integrated systems and have internal veterinary services (i.e., known as corporate veterinarians) who provide oversight of antimicrobial product distribution through company (or partner) feed mills and farms (i.e., as part of the coordinated supply chain, for example in the poultry sector, the hatcheries). The VCPR in the agrifood system is represented in the figure, indicating that a single farmer/producer may interact with different actors in the AMU distribution chain (e.g., corporate veterinary services or independent/private veterinary practice) regarding AMU decisions pertaining to the flock or herd.

### 2.2. Stakeholders’ Need for AMU Data

In the initial meeting of the working group, participants were asked for their organization’s specific primary objective for AMU surveillance and the utility of the associated data. There were eight food animal sectors and an allied industry (feed mill) that participated in the survey. Responses are shown in Box 1.

Box 1Participants’ responses when asked about their objectives for antimicrobial use surveillance. AMU—antimicrobial use; AMR—antimicrobial resistance.
**Primary objectives for antimicrobial use surveillance and the utility of the data collected:**
“*To garner public trust, to demonstrate social responsibility*”;“*To understand AMU (in my sector)*”;“*To provide AMU benchmarks (reference or targets) and trends that will inform decision making/policy and demonstrate the impacts of interventions*”;“*To provide oversight of AMU*”;“*To reduce reliance on antimicrobials to preserve antimicrobial efficacy*”;“*To improve/inform/demonstrate antimicrobial stewardship and prudent use*”;“*To provide international comparisons*”;“*To guide research priorities*”;“*To educate the public, veterinarians and producers about AMU/AMR*”.


The ability to demonstrate social responsibility was cited by the majority of participants as an overarching objective for the collection of AMU data. However, the ultimate outcome of AMU surveillance indicated was to preserve access to efficacious antimicrobials by strengthening stewardship. Some commodity groups indicated specific AMU surveillance objectives, for example, the need for AMU surveillance data to demonstrate the need for additional licensed antimicrobial options for sheep. Long-term outcomes identified included the ability to address certain AMU practices through the optimization and rationalization of use and to decrease the health burden associated with AMR. Other desired outcomes were related to the sustainability of their commodities, such as economic benefits, impact on production performance parameters, and return on investment. Based on the desired AMU objectives and outcomes, subsequent working group discussions for the inclusion of variables in the MDS-AMU-surv were guided by the AMU/AMR working group vision formed during the first meeting (i.e., in Ottawa, Ontario on 16–19 October 2016). The working group determined that “*AMU surveillance is necessary to inform decision making to objectively address AMU stewardship and to maintain public trust in the sustainable production of safe and affordable food from humanely raised animals*”. The working group considered that understanding the need for AMU surveillance could direct the next steps including the development of the MDS-AMU-surv.

### 2.3. Outputs Desired and Important Considerations When Reporting AMU Data

Several participants emphasized the need for detailed information to better understand AMU in animals. In addition to the ability to quantify AMU, the importance of context was noted (e.g., where—national/regional, who—the population at risk and size, when—time exposed, and why—reason for AMU). Some cautioned that to be successful the surveillance system must not increase the reporting burden on the producer. The group reviewed how existing surveillance platforms such as data from importers/manufacturers/compounders and end-users, as well as future data collection from prescription and dispensation levels, generate the outputs desired by the different sectors (Figure 2). 

### 2.4. The MDS-AMU-Surv Development, Framework for Data Collection, and Data Sources

#### 2.4.1. Core MDS-AMU-Surv Data Elements

A guiding principle for the development of the AMU surveillance capacity considered that there was a “*need for minimum data to tell a coherent story, cognizant of cost and effort*”. Exploring existing infrastructure to collect AMU data could optimize resources, again avoiding additional reporting requirements by producers. The working group recognized that national-level organizations (marketing boards or industry sectoral groups) could support existing farm-level data collection (e.g., CIPARS Farm AMU/AMR Surveillance or provincial AMU surveillance initiatives) vs. initiating competing, and possibly redundant, surveillance programs. There was no appetite for regulated use reporting similar to the Yellow Card or Differentiated Yellow Card System in place in other countries such as Denmark [19]. It is important to note that in the aquaculture sector (not yet providing data at the time of MDS-AMU-surv development in 2016–2017), the Aquaculture Reporting Regulations (Fisheries Act, Section 35–36) already required owners and operators of aquaculture establishments to report their annual antimicrobial deposit data [12,20]. There was interest in AMU data capture at the veterinary prescription/dispensing level of the distribution system (white boxes in Figure 1). The core elements of the MDS-AMU-surv are described in Table 1.

#### 2.4.2. Data Source

Several sector representatives indicated that various platforms (i.e., “*what records can you provide?*”) could be optimized for data collection that would fulfil the MDS-AMU-surv. Table 1 includes examples of data from existing records that could be used to collect the core MDS-AMU-surv elements and reporting and communications considerations (i.e., different ways to fulfil data requirements for the derivation of AMU metrics and indicators).

#### 2.4.3. AMU Surveillance Framework

As previously described, some sectors were interested in conducting AMU surveillance independent of CIPARS Farm AMU or CIPARS VASR components. Data collected by the industry could thus complement national/published information for the comprehensive assessment of the impact of AMU stewardship actions. These data could also be linked to economic or production variables or indicators not collected through either CIPARS or the VASR system: Frequency: An annual frequency of data collection was perceived as being sufficient for tracking AMU over time by the participants in most cases.Number of farms: depending on the surveillance objectives and capacity/resources, the sampling frame could be a census of farms across Canada, regional census (e.g., all farms in a given area), random sample (i.e., targets a certain percentage of farms across the area of interest), stratified sample, convenience/voluntary sample, those with electronic records, and those over a certain farm size.Who collects the data: existing platforms (government such as CIPARS or other organizations), provincial government, veterinary associations, producer associations, and processor associations were identified as options. These parties have direct access to some data or have a role in the AMU distribution chain (Figure 1).Incentives: existing data collection is either voluntary (e.g., CIPARS Farm) or regulatory (e.g., VASR, DFO) in nature. In private sector/industry initiatives, these are industry requirements as part of their on-farm food safety program, but there are no other incentives provided to the producer.

### 2.5. Application of MDS-AMU-Surv in Analysis and Reporting of Data

As shown in Figure 3, the MSD-AMU-surv core data elements enable the derivation of metrics for the reporting of some commonly used AMU indicators, including those that are count-based, weight-based, and dose-based, which have already been used in existing AMU surveillance systems in Canada [9] and elsewhere [24,25,26,27,28,29]. Figure 4 shows an example where complementary AMU surveillance components such as the CIPARS VASR and CIPARS Farm AMU Surveillance and other systems utilize the MDS-AMU-surv data elements for informing national AMU stewardship and a more complete picture of the status of AMU in the animal sector in Canada. In any surveillance program, data could be described using a nationally defined categorization system such as Health Canada’s Veterinary Drugs Directorate categorization of antimicrobials based on their importance to human medicine [30]. Surveillance data should enable the reporting of AMU aggregated at a national or regional/provincial level. Where possible, reporting by stages of production facilitates the identification of stages deemed to be high users of antimicrobials, and thus opportunities for intervention. The ability to report by primary or main indications for use and disease groups or specific diseases provides value to the collected data. It was recognized that capturing this information for quantitative AMU may be challenging because there are multiple uses for each antimicrobial. In this case, expert opinion may be obtained from participating veterinarians on the most likely reason for AMU. There were suggestions that animal health data such as mortality and occurrences of diseases should be collected to provide context to temporal trends; for example, spikes in total AMU or certain antimicrobials observed during a particular surveillance year may be due to a disease outbreak or emergence.

## 3. Discussion

The development of the MDS-AMU-surv in the Canadian food animal sector demonstrates that different sectors can work together under a voluntary platform. Multisectoral collaboration such as this is useful in addressing components of the Pan-Canadian Framework for action on AMU/AMR in the absence of a formal multisectoral One Health governance mechanism [32]. The authors acknowledge that the working group discussions on MDS-AMU-surv predate the regulatory shifts in AMU; this paper is reflective of the changes including the mandatory reporting requirements to the manufacturers, importers, and compounders, refinements in CIPARS surveillance methodology, and reporting (i.e., metrics and indicators), additional AMU data now provided by the aquaculture industry, and emerging initiatives. AMU monitoring and surveillance programs, whether these are government or industrial sector initiatives [24], are essential components of integrated AMU/AMR surveillance systems [33] and can contribute to national AMU/AMR data reporting and communication [24,34]. For example, food animal sectors in the United Kingdom provide data to the UK Veterinary Antimicrobial Resistance and Sales Surveillance Report [35], the Netherlands Veterinary Medicines Institute [36] program provides data to Monitoring of Antimicrobial Resistance and Antibiotic Usage in Animals in the Netherlands (MARAN) [37], and VetStat is the source of AMU data for the DANMAP of the Danish Programme for the surveillance of antimicrobial consumption and resistance [38]. Competent authorities provide oversight and coordination of AMU/AMR surveillance activities and are also responsible for reporting and communicating the data [24,34]. In various European countries, the livestock industry and other private sector organizations have developed their own AMU monitoring programs, which align with national AMU stewardship goals [24,34]. In addition to monitoring, these programs have set AMU reduction targets and provided benchmarking to promote behaviour change in AMU practices [19,36]. While there is no global consensus on which AMU indicator to use for reporting and for fulfilling various AMU surveillance objectives, existing measurements developed at the national level are used. For example, in Thailand, a set antimicrobial consumption reduction target in animals was based on mg/PCU_Thailand_ [39], and in the Netherlands, a reduction in AMU is measured using species-specific DDDA/animal-year [36]. 

Membership in CAHSS is voluntary and provides a forum for stakeholder engagement to address specific topic areas in animal health. The AMU/AMR working group membership includes the government, industry, academia, and other interested parties. Representation from various food animal sector is vital in addressing the shared AMR threat in the human–animal–environment interface [13]. In this circumstance, the federal government’s participation (Public Health Agency of Canada, Canadian Food Inspection Agency, Agriculture and Agrifood Canada, and Health Canada) provided the expertise regarding AMU surveillance program development, and in return, the government benefited from the information generated to inform policy and action. Industry representation provided practical considerations necessary for a sustainable AMU surveillance framework design. Those from academic institutions could use surveillance initiatives as platforms for research, while veterinary clinics or organizations could apply AMU monitoring principles in their own practice. The MDS-AMU-surv development was specific to AMU surveillance, but it could be used as a model for addressing other components of the global and national AMR action plans directed at the animal sector or larger stakeholder groups for addressing shared threats in the human–animal–environment interface. Examples of themes that could be discussed by the CAHSS AMU/AMR working group are infection prevention and control (biosecurity, animal health, and transition from AMU-dependent production systems to reduced AMU production systems) and AMU/AMR communication and advocacy. At the time of writing, the CAHSS AMU/AMR working group is exploring the utility of the MDS-AMU-surv for the development of AMU metrics and indicators. Collecting the relevant input parameters through MDS-AMU-surv is essential for describing the AMU information collected and possibly for the alignment of analysis, reporting, and communication with existing AMU surveillance systems.

Detailed data are required for characterizing AMU. The data collected through CIPARS-VASR provide information on total quantity sold for use in all animal species (with total quantities for eleven different animal species groups and aquatic animals) and the diversity and proportion of antimicrobial classes used. This system enables the detection of overall trends in sales and could be used to assess the impacts of regulatory or other changes on AMU at the national or provincial levels. The CIPARS Farm AMU data provide more detailed and specific information regarding AMU in the major food animal species and contextualize sales data such as species-specific AMU and classes used by sector and by disease type. The DFO data provide information on the annual deposit of drugs and pesticides; under the current regulations, each operator is required to report on a yearly basis the amount of drugs deposited, the dates, and the reason for use [12,20]. Farm-level surveillance thus captures other aspects of AMU that are potentially important to the understanding of AMR (e.g., average dose, length of exposure, and production stage). In this regard, the vast majority of countries in Europe also have active AMU monitoring independent of an existing national AMU surveillance program to fulfil other AMU objectives such as benchmarking or targeting AMU quantity reductions within the animal sector (or specific commodity) [24,25]. 

Recently, in an effort to enhance stewardship of AMU in Canada, additional new AMU monitoring programs have been developed that are independent from the long-standing CIPARS Farm surveillance and the newer CIPARS VASR component. Through a multisectoral collaboration, the province of Québec is developing an AMU surveillance program with the aim of collecting data across the major food animal species raised in the province [40]. The Canadian Veterinary Medical Association is also supporting a veterinary practice AMU data collection system as part of the Stewardship of Antimicrobials by the Veterinarians Initiative (SAVI) [41]. Veterinary prescribing and dispensing data are currently unavailable, which are identified as Level 2 in the Canadian AMU distribution chain (Figure 1). The SAVI pilot project aims to address this AMU data gap. It was recognized that VCPR are complex in some sectors (interactions between multiple actors/players) and that the data may lack the granularity required for understanding the major factors affecting AMU. Methods for data collection at this level should consider potential double reporting (i.e., multiple veterinarians providing oversight to the same establishment or company) when the aim is to quantify AMU, and to contextualize it to the appropriate biomass. Additionally, in recent years, various poultry sectors in Canada have also developed their own AMU data collection as part of their on-farm food safety program and to monitor the progress of their AMU stewardship actions [42,43]. At the industry level, existing platforms in Canada, such as the flock sheets reporting or farm flock health records as part of the poultry on-farm food safety program, are now utilized to collect AMU data. Although the data collected may not be as comprehensive as the CIPARS Farm AMU Surveillance, this generates information (e.g., frequency of use) complementary to the CIPARS Farm AMU Surveillance program. At the time of writing, the methodology for the quantification of the AMU collected is under development. Thus, industry-led programs could progressively be built upon to generate quantitative data (i.e., possibly larger coverage such as census vs. sample of farms) enabling much better comparability with existing national or global surveillance programs. 

The MDS-AMU-surv proposed in this paper is expected to be sufficient to generate basic AMU metrics and indicators in any livestock sector. CIPARS has advanced its farm-level analysis by developing the defined daily dose in animals using Canadian standards (DDDvetCA) [10,44]. Dose-based indicators were applied to the CIPARS Farm AMU Surveillance data for characterizing AMU trends between species and investigated for their utility for studying AMU–AMR linkages [45]. Additionally, the dose-based indicators were used for monitoring the impact of the changes in voluntary and regulatory AMU practices (i.e., from disease prevention to disease treatment or disease control, progressive elimination of certain classes of antimicrobials) [45,46]. Other sectors intending to develop their capacity for AMU surveillance could progressively improve their data collection and analytic capacity to enable the reporting of indicators relevant to their sector. While there may be different AMU metrics and indicators preferred to inform AMU stewardship in the different sectors, the collection of the MDS-AMU-surv will meet the needs of the government and industry (inform policy, risk assessment, and monitoring of the impact of AMR interventions on the animal–environment–human interface). The MDS-AMU-surv could be progressively improved over time to align with global reporting requirements and the future harmonization of integrated surveillance methods.

## 4. Materials and Methods

### 4.1. The CAHSS Working Group

A multistakeholder workshop was held in Ottawa between October 16 and 19, 2016 to gain a better understanding of producer and animal industry needs for AMU surveillance and to define areas where producers and producer associations would find value in collaborating on AMU surveillance. Participants consisted of primarily industry associations, veterinary groups, and government agencies engaged in AMU surveillance in food animals. Following this initial meeting, core CAHSS members (i.e., from the Public Health Agency of Canada, Canadian Food Inspection Agency, Agriculture and Agrifood Canada, and Health) brought together representatives from the government (expertise in public health, veterinary medicine, surveillance, epidemiology, and inspection systems), academia, various livestock animal sectors/marketing boards, and allied industry representatives (e.g., feed mills and pharmaceutical industries) and veterinary associations that had interest in AMU/AMR surveillance, research, and policy development. These representatives ultimately formed the CAHSS AMU/AMR working group. The working group convened in meetings between January 2017 and December 2018 to discuss the development of the minimum AMU dataset. An average of 25 (ranging from 21 to 35) representatives from 24 organizations participated in the quarterly meetings. 

### 4.2. Review of Existing AMU Surveillance Infrastructures and the Antimicrobial Supply Distribution Pathway

In 2017, the government of Canada published the Pan-Canadian Framework for Action on AMU/AMR [5], which aligns with the Global Action Plan on AMR [1]. The MDS-AMU-surv dataset development contributes to the enhancement of the surveillance and stewardship components of the Pan-Canadian Framework. In this regard, several parties, including private sectors, were interested in documenting AMU practices in their sector. The CAHSS AMU/AMR working group recognized the need for a cohesive method for AMU data collection. A report titled “*Non-human antimicrobial use surveillance in Canada: Surveillance Objectives and Options*” was prepared by a Committee under the Council of Chief Veterinary Officers in Canada, which focused on AMU in food-producing animals [14]. Part of this report reviewed earlier versions of Canadian (i.e., CIPARS) and international (e.g., Denmark [38], the Netherlands [37], Sweden [47], and UK VARSS [35]) AMU/AMR surveillance programs and assessed various options in a Canadian context, including potential challenges for data collection (e.g., regulatory vs. voluntary mechanisms for AMU reporting or data collection; availability of resources and industry uptake/interests). 

### 4.3. Survey of CAHSS AMU/AMR WG to Understand the AMU Data Needs 

A survey was sent to the working group to better understand the objectives of participating organizations regarding AMU surveillance, desired outputs, and the feasibility of collecting specific data elements. The survey comprised 11 sections (Appendix A). Responses were collated in Microsoft Excel (Professional Plus 2016), and scores were summarized based on:Data inclusion scores: 0—not required, 1—nice to have, and 2—must be included.Data collection feasibility scores: 0—not feasible, 1—could start collecting with significant effort, 2—could start collecting with minimal effort, and 3—currently being collected by your organization.

An informal thematic analysis was conducted based on the participating organization’s response to the survey when asked about the specific AMU surveillance objectives and the intended utility of the data collected. 

### 4.4. Development of the MDS-AMU-Surv

The intent of the MDS-AMU-surv development was not to create an exhaustive list of variables to be used as a standard by each of the sectors with an interest in collecting AMU data. Instead, the primary consideration was to identify core data elements that could be collected in a number of different ways according to their capacity and capabilities within each commodity/sector and would provide information necessary for development of policies or intervention. Ultimately, the metrics and indicators generated would potentially align with existing national (e.g., CIPARS) and international AMU surveillance programs, for example, the OIE and the European Surveillance for Veterinary Antimicrobial Consumption [28,29], with which industries’ or any interested parties’ AMU data could be compared. 

The list of variables suggested were intended for generating quantitative AMU measurements but could also generate qualitative (count-based) AMU measurements. Obtaining the contextualizing animal demographic data enables the estimation of the most commonly used AMU metrics (e.g., mg/kg animal biomass or mg/population correction unit (PCU)) that could potentially evolve into more advanced measurements (e.g., dose-based indicators) or other relevant indicators for describing AMU.

In aquatic animals, the reporting platform was examined. The owners/licensed operator under the AAR of the Fisheries Act reports antimicrobials deposited to the DFO annually (i.e., prescribed and dispensed for use in aquaculture animals). The reporting platform (Microsoft Excel, Professional Plus 2016) comprises variables largely similar to the MDS-AMU-surv, but animal biomass data have not yet been collected. A link to the current template can be found in Appendix B. At the time of writing, this template is being updated to reflect the changes in the Aquaculture Activities Regulation.

## 5. Conclusions

The development of the MDS-AMU-surv in the Canadian food animal sector demonstrates that different sectors can work together under a voluntary platform and a collaborative, consensus-building process could be used in future activities related to the Pan-Canadian Framework for Action on AMR.

## Figures and Tables

**Figure 2 antibiotics-11-00226-f002:**
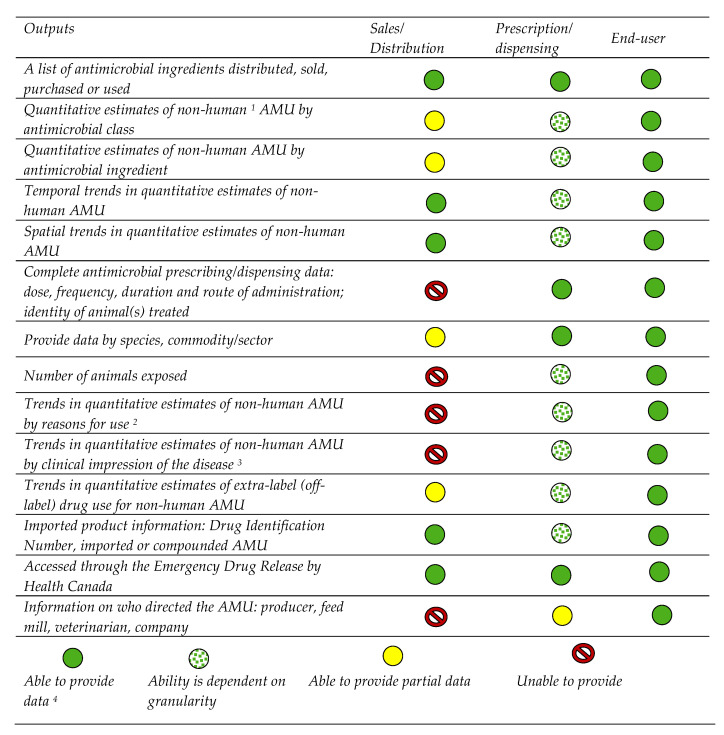
Antimicrobial use outputs from existing or future data collection points in the antimicrobial supply distribution pathway (figure modified from Reference [14]). ^1^ Nonhuman antimicrobial use in the table pertains only to animals. ^2^ Veterinary medical use such as disease treatment, control/metaphylaxis, prevention/prophylaxis, and additionally growth promotion. ^3^ Clinical impression by the veterinarian with input from producers and postmortem findings with or without laboratory diagnoses. ^4^ Ability to provide data (green circles) may be different for data collected in certain situations (i.e., confidential business information). AMU—antimicrobial use.

**Figure 3 antibiotics-11-00226-f003:**
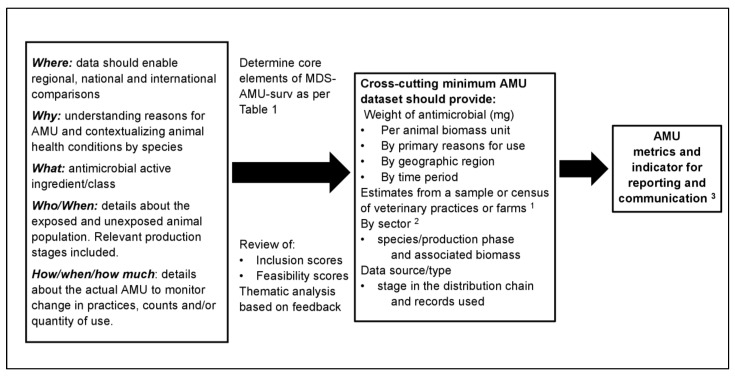
Summary of the considerations for the development of minimum dataset for antimicrobial use surveillance (MDS-AMU-surv), core data elements, outputs, and application. AMU—antimicrobial use. ^1^ CIPARS program collects information from sentinel farms (pigs, broiler chickens, turkeys, feedlot beef, dairy, and layers). ^2^ Animal biomass information required for AMU indicator estimations. ^3^ AMUmetrics and indicators collected are based on the organization/sector’s AMU surveillance or study objectives and the stage of the development of the AMU surveillance program (i.e., during the early implementation stages, core elements may be incomplete). In aquaculture (data accessed by CIPARS under open data policy), Aquaculture Activities Regulations (under the Fisheries Act) require all operators/farms to report deposited antimicrobials (i.e., the quantity of antimicrobials prescribed and dispensed for use in aquaculture).

**Figure 4 antibiotics-11-00226-f004:**
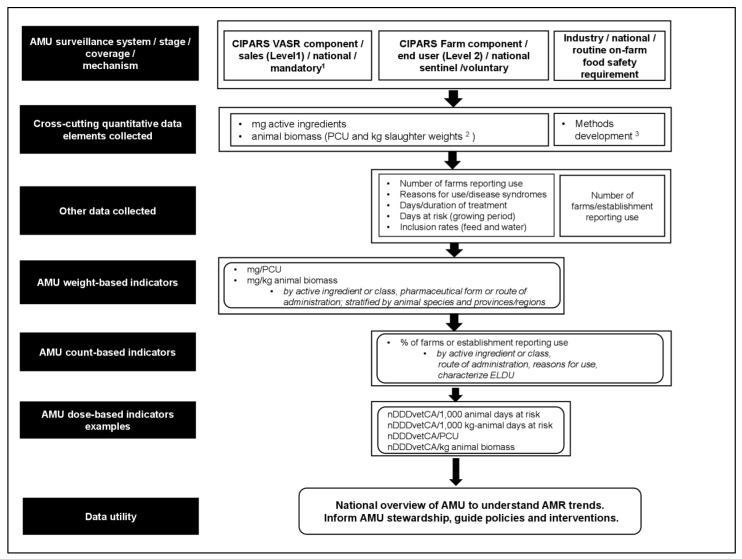
Application of the MDS-AMU-surv in Canadian animal sector. ^1^ In addition to VASR, CIPARS access open data collected by Fisheries and Oceans Canada under the Aquaculture Activities Regulations of the Fisheries Act. ^2^ Aquaculture data are not yet reported by biomass unit (i.e., using the AMU indicators routinely used by CIPARS for AMU reporting in terrestrial animals), but exploration of methodological options for data analysis and future reporting are underway [31]. ^3^ At the time of writing, the industry (i.e., poultry sector) utilizes count-based indicators to communicate results within their producer networks; as mentioned above, methodology for quantitative AMU indicator analysis and reporting are underway. AMU—antimicrobial use. ELDU—extra-label (off-label) drug use. nDDDvetCA—number of defined daily doses in animals using Canadian standards. PCU—population correction unit. VASR—Veterinary Antimicrobial Sales Reporting system; the data collection platform for national sales data. Levels 1 and 2 pertain to the stage of data collection indicated in Figure 1.

**Table 1 antibiotics-11-00226-t001:** Core elements of minimum dataset for antimicrobial use surveillance and considerations for inclusion.

Core Data Elements	Examples of Data That Would Fulfil the Element	Considerations for Surveillance Communication and Reporting
Antimicrobial active ingredient	On-farm treatment records (e.g., flock sheets), Farm purchases, Feed sales, Veterinarian’s medical records/prescriptions, Veterinarian’s dispensing record, Veterinary purchases, Manufacturer sales	Available in a number of formats, including the product or trade name, or the antimicrobial active ingredient itself. These data can then be easily converted to amount of active ingredient using a glossary of products that provides the concentration of active ingredient (and Anatomical Taxonomic Index codes for veterinary antimicrobials (ATCvet code)). This glossary should be linked to current references (i.e., to be used as a common reference across surveillance programs): Compendium of Medicating Ingredients Brochures [21] and the Drug Products Database [22]; Pharmaceutical products (with DIN) and compounded products (without DIN) as reported in the VASR system; Compendium of Veterinary Products [23]. **Weight of antimicrobials** (i.e., numerator data in some indicator calculations): pertains to the total quantity of AMU across all routes of administration. This is important for analysis of trends and the basis for further quantification of AMU (dose-based indicators).
Biomass unit	On-farm records (e.g., flock sheets), Processor records, Census data (to geographic region), Expert consensus (e.g., proportion of total animals in each production level, multiplier values for biomass calculation (kg) ^1^)	Required for both numerator (animals treated) and denominator (total animals) estimates. Total population (including mortalities and number of animals introduced to the flock or herd) and weights are important to collect but can come from several different sources, including: Live pre-slaughter weights; Slaughter weights; Actual average treatment weights. In certain sectors, the inclusion of multiple weight and age groups and relevant production class was viewed as useful additional information ^2^, examples: Nursery pigs, grow–finisher pigs; Veal calves, feedlot beef; Broilers, broiler breeders, layers, layer breeders **Animal biomass** (i.e., denominator data in some indicator estimations composed of the exposed and unexposed groups): pertains to the total number of animals presumably exposed to antimicrobials multiplied by an appropriate multiplier value (kg weight). These data enable comparison of AMU over time and between species.
Reasons for use	Expert consensus (by proportion of total AMU) ^3^, On-farm records, Veterinary medical records	The ability to capture the main indications for medical AMU, such as for disease prevention/prophylaxis, disease control/metaphylaxis and disease treatment, or growth promotion, was viewed as necessary. More specific reason for using data (e.g., respiratory vs. gastrointestinal vs. other disease treatment) was also desirable but not considered a core necessity. This is additional information for contextualizing AMU and AMR data and the impact of regulatory changes in AMU.
Geographical location	On-farm records, Veterinary records, Processor records, Hatchery delivery receipts	Collecting the province or the region where the animals are raised or antimicrobials sold (for the VASR system) is important to enable geographical comparisons of use.
Time component	Multiple years, Yearly, Monthly, Weekly, Daily, Real-time, Production cycle/s (specify)	Required to compare trends over time and gauge changes in AMU following interventions. There are various time elements that need to be captured for more advanced quantification and analysis [24,25,26,27]: When (age of the animals) animals were likely exposed; Duration of treatment for each antimicrobial administered (i.e., if available, for one full water treatment course, medicated ration, or total days exposed to all antimicrobials); Total days at risk (i.e., which is equivalent to the duration of the growing cycle, needed in some AMU indicators such as TI_1000_ or TI_100_ ^4^); Data coverage (e.g., one growing cycle or annually).

^1^ Estimates from producers (and producer/farmer associations or marketing boards), researchers, and veterinarians. ^2^ Studies for identifying the production phase with highest use of antimicrobials, which may be linked to diseases prevalent during that stage. ^3^ Expert opinion from species-specific veterinarians (derived from their veterinary prescription and animal health records). ^4^ TI_1000_ or TI_100_—Treatment Incidence 1000 or 100 (CIPARS used the term) is number of defined daily doses in animals using Canadian standards per 1000 or 100 animal days at risk.

## Data Availability

Not applicable.

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
