# Peer review of "Canadian Collaboration to Identify a Minimum Dataset for Antimicrobial Use Surveillance for Policy and Intervention Development across Food Animal Sectors"

_antibiotics, 2022, doi:10.3390/antibiotics11020226_

Round 1

Reviewer 1 Report

Generally, a beautifully written article. However very strange section in the Introduction (lines 50-54) in which grammar errors occur and repeats of the same sentence occur. These have been noted in the attached document along with other minor grammar issues throughout.
Notably, the supplementary material/appendices are named differently several times - needs correcting in both main text and supp material to avoid confusion. Additionally, appendix C/A3 should be given as downloadable document - not as a link which may expire.
An encouraging paper promoting collaboration and data sharing in an important field, although not particularly impactful. Notwithstanding, if successful I think it has the potential to be impactful in future. 

Author Response

Dear Reviewer,

We thank you for your thorough review and recommendations to improve our manuscript.  We addressed your comments and corrected errors identified during your review.

The Supplementary Materials have been re-labelled and cross-referenced throughout the manuscript. Below are the responses to the line items you have indicated.

Sincerely,

The authors

Reviewer 1 – responses to specific line items.

Generally, a beautifully written article. However very strange section in the Introduction (lines 50-54) in which grammar errors occur and repeats of the same sentence occur. These have been noted in the attached document along with other minor grammar issues throughout.
Notably, the supplementary material/appendices are named differently several times - needs correcting in both main text and supp material to avoid confusion. Additionally, appendix C/A3 should be given as downloadable document - not as a link which may expire.
An encouraging paper promoting collaboration and data sharing in an important field, although not particularly impactful. Notwithstanding, if successful I think it has the potential to be impactful in future. 

On behalf of all the authors, many thanks for your very positive feedback, for detecting the errors, and providing recommended changes/edits. We hope that this manuscript will be of value to the readership of Antibiotics and could be a good reference for countries that are developing their capacity for antimicrobial use surveillance.

Main manuscript

  • Line 45: extra “)” bracket

Response: Thank you. Error has been corrected.

  • Line 50-1 and 54-55 (and elsewhere): Should the entry in speech marks “Tackling Antimicrobial Resistance and Antimicrobial Use: A Pan-Canadian Framework for Action” be italicized (As it is here: https://pubmed.ncbi.nlm.nih.gov/29770049/)

Response: Thank you for this comment. We harmonized the format with the published document.

  • Line 52: Speech mark grammar – “is

Response: corrected.

  • Line 53: Repeated phrase – “generating knowledge through antimicrobial use (AMU) and AMR 53 surveillance”

Response: the statement has been revised accordingly.

  • Line 50-54: In fact, this whole section does not make sense? It repeats different sentences. Should be rewritten

Response: We thank you for this comment; we simplified the sentence to enhance readability.

  • Line 231: grammar error “participat-ed”

Response: thank you for detecting the error (it occurs during the formatting of the bibliography); this was modified accordingly.

  • Line 231: change colon to period “Responses are shown in Textbox 1.”

Response: modified accordingly.

  • Line 251: double bracket again “))”

Response: modified accordingly.

  • Line 251: Space after speech mark/before AMU - that “AMU surveillance

Response: modified accordingly.

  • Line 258: grammar error “tobetter”

Response: modified accordingly.

  • Line 266-267: Don’t need phrase “Nonhuman uses in Figure 2 pertain only to animals and excludes crops.” as this is specified in the figure caption

Response: thank you for this comment. The phrase has been excluded to avoid redundancies.

  • Line 267-268: Missing line gap in subsections

Response: one line added to separate the last paragraph from the next subsection.

  • Line 273-274: quotation should be in italics

Response: modified accordingly.

  • Line 290-291: quotation should be in italics

Response: modified accordingly.

  • Line 297- (Figure 2): Nice figure but remove “antimicrobial use (AMU)” from outputs column. Give acronyms in figure caption (just for consistency)

Response: modified accordingly.

  • Line 306: missing full stop.

Response: modified accordingly.

  • Line 309: Add footnote number 4? Similarly put 4 superscript with “TI1000 or TI100” in table 1

Response: thank you for this suggestion; modified accordingly.

  • Line 336 - (Figure 3): Missing explanation of acronyms – figure and legend must be free-standing

Response: we thought this could be figure 4. We also added a table of acronyms located before the references

  • Line 417: text “eleven” rather than number

Response: modified accordingly.

  • Ine 505 - “Supplementary Materials I” – I assume this is Appendix 1 – Table 1. Please make clear

Response. Modified accordingly

  • Lines 566-569: “Appendix A,B,C” aren’t they 1,2,3?? Needs correcting

Response. Modified accordingly

  • Line 569: “Appendic” spelling mistake

Response. Modified accordingly

  • Line 572-574: Appendix 3 should not be given as a link. It should be in a downloadable spreadsheet format. Using a link is not reliable.

Response. Thank you for this comment.  The spreadsheet will be added as Appendix B.

Supplementary/appendix material

  • Need consistency in naming of the supplementary materials throughout manuscript e.g. Line 505 “Supplementary Materials I” – I assume this is Appendix 1 – Table 1
  • Additionally in the manuscript they are calle Appendix A/B/C however in supplementary material they are called “ Table A1”
  • Can be confusing needs correcting

Response: thank you for these observations. We fixed the Supplementary Materials, relabelled the Tables and ensured they are cited properly within the text. Upon consultation with the Fisheries and Oceans, we have also provided the aquaculture data collection template as Appendix B.

Reviewer 2 Report

I am delighted to review this manuscript, covering an important aspect of the subject, with a good presentation of results, making this article interested to the readers of the journal Antibiotics, though the author could have concise the discussion. The manuscript follows the scope of the journal Antibiotics.

 I would recommend this manuscript could be published in Antibiotics in the current form.

The authors need to address the below-mentioned queries.

1. Introduction is a bit lengthy, could be concise and to the point.

2. The author could mention the AMU monitoring and surveillance programs associated with other continents as well.

3. The author could comment on the global AMU surveillance programs a bit.

Author Response

Dear Reviewer,

On behalf of all authors, many thanks for your very positive feedback and recommended changes/edits. We hope that this manuscript will be of value to the readership of Antibiotics and could be a good reference for countries that are developing their capacity for antimicrobial use surveillance.

Sincerely,

The authors

Reviewer #2 - responses to specific line items.

I am delighted to review this manuscript, covering an important aspect of the subject, with a good presentation of results, making this article interested to the readers of the journal Antibiotics, though the author could have concise the discussion. The manuscript follows the scope of the journal Antibiotics.

 I would recommend this manuscript could be published in Antibiotics in the current form.

The authors need to address the below-mentioned queries.

  1. Introduction is a bit lengthy, could be concise and to the point.

Response: thank you for this comment; we have truncated the introduction but emphasized the collaboration leading to the development of the MDS-AMU surv (Lines 127-129). Miscellaneous information (2 paragraphs on AMU surveillance systems) were moved to the discussion section which would address, in part, your comments below.

  1. The author could mention the AMU monitoring and surveillance programs associated with other continents as well.

Response: thank you for this comment. The surveillance programs in the EU were mentioned throughout as these are the most established and have advanced methodology for estimation. An example from Asia (Thailand, reference #39) was mentioned. The discussions on the international surveillance systems, their utility for AMU stewardship and addressing various AMU objectives, and the lack of a global consensus on which AMU indicator to use are grouped together in lines 420-440 of the discussions (modified from the introduction section).

  1. The author could comment on the global AMU surveillance programs a bit.

Response: well noted, to complement the statements in lines 420-440, on lines 507-509, it is stated “industry-led programs could progressively be built upon to generate quantitative data (i.e., possibly larger coverage such as census vs. sample of farms) enabling much better comparability with existing “national or global surveillance programs”. Additionally, we added a statement on lines 535-537, indicating that “ The MDS-AMU surv could be progressively improved over time to align with global reporting requirements and future harmonization of integrated surveillance methods.”
